# GRID: Scalable Task-Agnostic Prompt-Based Continual Learning for Language Models

## Abstract

Prompt-based continual learning (CL) provides a parameter-efficient approach for adapting large language models (LLMs) across task sequences. However, most existing methods rely on task-aware inference and maintain a growing set of task-specific prompts, which introduces two major challenges: (1) severe performance degradation on earlier tasks under task-agnostic inference, and (2) limited scalability due to prompt memory accumulation as task sequences grow. In this paper, we present GRID, a unified framework designed to address these challenges. GRID incorporates a decoding mechanism that enhances backward transfer by leveraging representative inputs, automatic task identification, and constrained decoding. Furthermore, it employs a gradient-guided prompt selection strategy to compress less informative prompts into a single aggregated representation, ensuring scalable and memory-efficient continual learning. Extensive experiments on long-sequence and negative transfer benchmarks show that GRID improves average accuracy and backward transfer, achieves competitive forward transfer, and substantially reduces prompt memory usage.

## 1 Introduction

Continual learning (CL) (Van de Ven & Tolias, 2019) enables models to learn from a sequence of tasks without retraining from scratch. CL systems build on prior knowledge while adapting to new tasks, which is crucial in dynamic real-world settings. Recent advancements, especially in NLP (Wang et al., 2024; Satapara & Srijith, 2024), have focused on three paradigms: *regularization-based methods* (Li & Hoiem, 2017; Kirkpatrick et al., 2017), *rehearsal-based methods* (Rebuffi et al., 2017; Sun et al., 2019), and *architecture-based methods* (Veniat et al., 2020; Douillard et al., 2022). While rehearsal-based methods are effective, they are impractical in privacy-sensitive scenarios. With the increasing complexity of pretrained models (Wang et al., 2023a), full model finetuning has become infeasible. This has led to the rise of parameter-efficient finetuning (PEFT) techniques (Ding et al., 2023), such as *prompt tuning* (PT) (Lester et al., 2021), which adapts large models by training only a small set of soft prompts, requiring less than 0.01% of the model's parameters.

Building on this, continual prompt tuning (CPT) extends PT to the CL setting by enabling models to learn task-specific prompts sequentially without modifying the base model (Wang et al., 2022b). Recent advancements in prompt tuning-based continual learning address key challenges like catastrophic forgetting and forward knowledge transfer. Wang et al. (2022a) introduced a dual prompt framework with a shared and task-specific prompt, though this approach faces limitations in retaining knowledge from past tasks. To mitigate these issues, ProgPrompt (Razdaibiedina et al., 2023) maintains a list of prompts for each task, progressively adding new ones while retaining old ones. SHLPT (Wu et al., 2024) extended this by proposing an attention-based similarity estimator to compose similar prompts for initialization while contrastively regularizing against dissimilar tasks, mitigating negative transfer.

Although prompt-based CL has received significant attention, two main challenges remain:

- First, while major prompt-based CL methods such as Progressive Prompts (Razdaibiedina et al., 2023), Q-Tuning (Guo et al., 2024), and SHLPT (Wu et al., 2024) report zero forgetting, thereby fully preserving prior knowledge, they depend on *task-aware inference*, where task identities are explicitly provided to retrieve the correct prompt for each previous task. In practice, however, task IDs are often unavailable during inference (Wang et al., 2024; Liang & Li, 2023).

- Second, most prompt-based CL methods struggle to scale efficiently in both time and memory as the prompt queue expands. For example, Progressive Prompts (Razdaibiedina et al., 2023) and SHLPT (Wu et al., 2024) assign a dedicated soft prompt to each task, leading to memory usage that grows linearly with the number of tasks. More recent approaches attempt to mitigate this issue by using PCA-based eviction (Guo et al., 2024) or by continually updating the same set of prompts over time (Wang et al., 2022b). However, these solutions either incur significant computational overhead (e.g., repeated SVD during eviction) or suffer from redundant prompt accumulation due to the lack of effective pruning or merging mechanisms.

The above challenges motivate us to investigate an important but underexplored task-agnostic scenario:

*The task identity is unknown at inference time, and the prompt pool has a fixed capacity. In this setting, the model cannot (1) guarantee zero forgetting by relying on task-specific prompts, nor (2) indefinitely store all prompts as the task sequence grows. Consequently, the model must perform prediction using the entire available prompt pool, while prompt pruning or merging becomes necessary to maintain the prediction quality as well as the bounded size of the pool.*

This paper focuses on the above setting and makes the following specific contributions.

- We observe that existing prompt-based continual learning approaches often struggle in task-agnostic scenarios. Our experiments reveal that when task identities are unavailable at inference time, performance on earlier tasks degrades substantially after training on new ones, with models frequently producing incorrect or ambiguous outputs. This issue is especially pronounced for encoder–decoder architectures, which generate labels as free text: without explicit task cues, the model may output label words from unrelated tasks encountered during pretraining or earlier learning. We refer to this phenomenon as *latent forgetting*, denoting the degradation in performance on earlier tasks under task-agnostic evaluation. Similar observations have been reported in prior work (e.g., (Guo et al., 2024)), showing that forward transfer often persists while backward transfer suffers severely in the task-agnostic setting.
- We introduce **GRID**, a unified framework that integrates constrained decoding with a principled gradient-guided prompt selection to address the above limitations:
  - A *decoding mechanism* that leverages representative inputs and constrained decoding, an approach applied to control the output space of pre-trained LLMs, to ensure label consistency and improve backward transfer (BWT) without relying on task IDs.
  - A *gradient-guided prompt selection* strategy that dynamically evaluates prompt usefulness and merges less informative prompts, significantly reducing memory usage while maintaining both forward and backward transfer performance.
- We conduct extensive experiments across long-sequence and negative transfer benchmarks. GRID improves BWT by up to 54%, reduces the number of forgotten tasks by 80%, and consistently matches or outperforms average accuracy of the state-of-the-art prompt-based CL baselines such as ProgPrompt and SHLPT under the task-agnostic conditions.

## 2 BACKGROUND AND CHALLENGES

### 2.1 PROBLEM SETUP

We consider a continual learning (CL) setting in which a model encounters a sequence of $N$ tasks $\mathcal{T} = \{T_1, T_2, \ldots, T_N\}$, where each task $T_i$ is associated with a labeled dataset $D_i = \{(x_j, y_j)\}_{j=1}^{|D_i|}$. Here, $x_j$ denotes an input instance and $y_j \in \mathcal{Y}_i$ is the corresponding label. The model is built upon a pretrained encoder-decoder language model $f(\cdot; \theta)$, whose parameters $\theta$ are kept fixed throughout learning. Rather than finetuning $\theta$, we adapt the model to each task $T_i$ by learning a soft prompt $\mathbf{p}_i \in \mathbb{R}^{l \times d}$, where $l$ denotes the prompt length and $d$ the embedding dimension. After observing tasks $\{T_1, \ldots, T_{t-1}\}$, we maintain a pool of learned prompts $\mathcal{P} = \{\mathbf{p}_1, \ldots, \mathbf{p}_{t-1}\}$, which serves as a memory of past task adaptations.

When a new task $T_t$ arrives, we initialize a new prompt $\mathbf{p}_t$ and train it using data from $D_t$, concatenated with the existing prompt queue $\mathcal{P}$. The backbone model $f(\cdot; \theta)$ remains frozen during training; only the new prompt $\mathbf{p}_t$ is updated. Let $\mathcal{P}^{(t)} = \mathcal{P} \cup \{\mathbf{p}_t\}$ denote the prompt configuration used during training for task $T_t$. The model prediction is then given by $\hat{y} = f(x; \mathcal{P}^{(t)})$. The training objective is to minimize: $\mathcal{L}_t = \mathbb{E}_{(x,y) \sim D_t}[\ell(f(x; \mathcal{P}^{(t)}), y)]$, where $\ell(\cdot)$ is the token-level cross-entropy loss.

## 2.2 Task-Agnostic Inference Setting

Building on the aforementioned motivation, this paper focuses on the following task-agnostic inference setting:

**Definition 1** (Task-Agnostic Inference). *The task identity is unknown at inference time. As a result, the model cannot rely on task-specific prompt selection. Instead, it performs prediction using the entire available prompt pool $\mathcal{P} = \{\mathbf{p}_1, \ldots, \mathbf{p}_{t-1}\}$, which may optionally include an aggregated prompt derived from previous filtering stages. The concatenated prompt sequence is prepended to the input and passed to the model. Formally, the prediction is defined as: $\hat{y} = f(x; \mathcal{P})$.*

The task-agnostic evaluation objective becomes: $\mathcal{L}_{\text{TA}} = \sum_{T_k \in \mathcal{T}_{\text{past}}} \mathbb{E}_{(x,y) \sim D_k} \left[ \ell(f(x; \mathcal{P}), y) \right]$. This formulation evaluates how well the concatenated prompt pool enables generalization to earlier tasks without needing to retrieve or know their corresponding individual prompts. This setting appears in two important CL scenarios: 1) *Online learning*, where a new task arrives and the model must evaluate or detect alignment with previously seen tasks (Aljundi et al., 2019a). 2) *Retroactive evaluation*, where a model trained on a task sequence is later evaluated on older tasks without access to their individual prompts (Chaudhry et al., 2018; de Masson D'Autume et al., 2019).

## 2.3 Challenges in Existing Prompt-based CL in Task-agnostic Setting

In this section, we discuss several challenges of existing methods in the task-agnostic case.

**Label Drift and Hallucination.** In task-agnostic settings, encoder–decoder models such as T5 generate free-form text outputs without explicit constraints, which often leads to two types of errors. The first is *label drift*, where the model produces semantically related but incorrect labels or syntactically invalid variants. The second is *hallucination*, where the decoder generates unseen or spurious labels due to the likelihood mass fragmenting over an expanding vocabulary. Qualitative examples of these failure cases are provided in Table 1.

Table 1: Qualitative comparison under task-agnostic inference. Blue indicates *label drift*, where Progressive Prompts predict syntactically incorrect labels. Red highlights *hallucination*, where the model generates invalid outputs not part of the task's label set.

| Task | Input Text | Label | ProgPrompt |
|------|-----------|-------|-----------|
| BoolQ | Did you ever lecture at Harvard? | true | true |
| BoolQ | The imperialist nation wanted to strangle the economy of its colony. | false | <pad>Fal |
| BoolQ | The neoclassical canon was rooted in traditional European aesthetics. | true | <pad>Fal |
| MNLI | Some of the buildings around the city square … colonial period. | entailment | <pad>True  |
| MNLI | The U.S. Army acceded … to keep U.S. forces in place. | entailment | <pad>4.0 |

**Inconsistent Label Mappings.** Prior approaches such as Progressive Prompts (Razdaibiedina et al., 2023) and SHLPT (Wu et al., 2024) typically rely on human-specified mappings (e.g., 0 → "negative") to convert numeric labels into text. Since these mappings can vary across datasets, they may introduce semantic inconsistencies that make cross-task generalization more challenging. Illustrative examples are provided in the Appendix (Table 19), showing how such differences can affect model behavior.

**Lack of Task Awareness.** Current methods often overlook the underlying task structure. As a result, even semantically related tasks (e.g., sentiment classification across different datasets) may be treated separately, which can limit opportunities for generalization and shared label alignment. This may lead to overlapping prompt representations and reduced transfer.

**Prompt Growth.** Under task-agnostic evaluation, the model makes predictions based on the entire prompt pool. Without mechanisms for pruning or merging, the prompt memory can gradually increase with possibly redundant prompt accumulation, which is not explicitly addressed in some existing prompt-based CL approaches such as Progressive Prompts (Razdaibiedina et al., 2023).

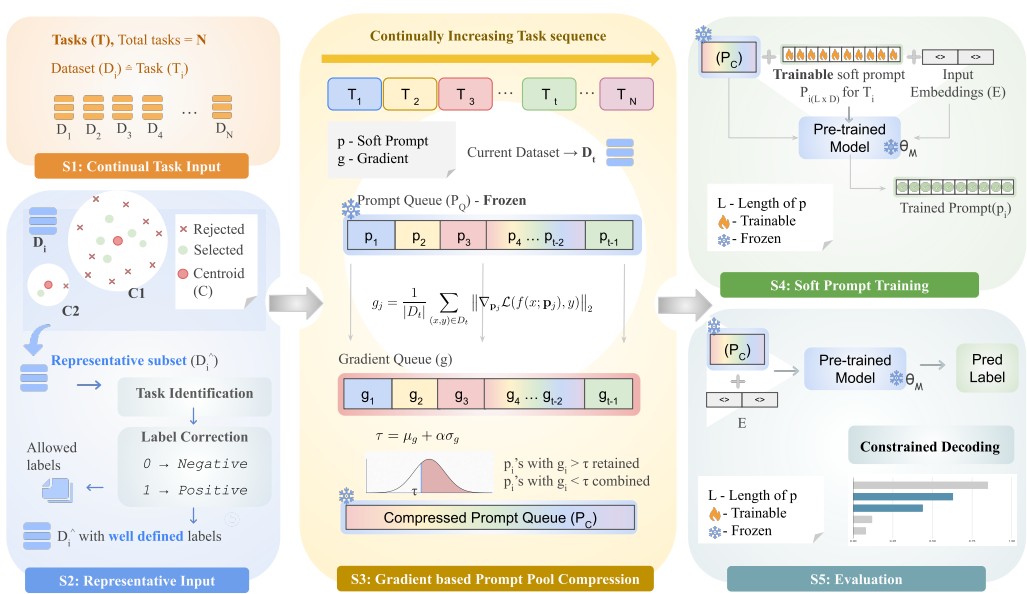

Figure 1: Overview of the proposed **GRID** framework. (S1) model receives a stream of tasks with corresponding datasets. (S2) Representative samples are selected via clustering for each task, and task identification is performed to ensure consistent label formats. (S3) Gradient-based prompt selection is applied: prompts from the frozen prompt pool are ranked based on their gradient norms with respect to the current task; (S4) compressed prompt pool is used to train soft prompts for new tasks with the base model frozen. (S5) During inference, constrained decoding ensures predictions are aligned with the identified task semantics.

These challenges highlight the need for a more structured and scalable framework for prompt-based continual learning under task-agnostic conditions.

## 3 METHOD

To address the aforementioned challenges, we propose a unified framework called **GRID**: **G**radient-based prompt selection with **R**epresentative sample selection, task **I**dentification, and constrained **D**ecoding. GRID integrates two complementary components: 1) An input pipeline that enhances backward retention and output consistency by selecting representative samples, performing task identification, and applying constrained decoding; 2) A gradient-based prompt scoring mechanism that reduces prompt pool size by identifying and merging less informative prompts while preserving relevant task knowledge. Figure 1 for the pipeline and Appendix A for full algorithms.

### 3.1 INPUT CONSTRUCTION FOR STABLE DECODING

Encoder-decoder models generate output labels as textual sequences. When task identity is unavailable at inference, unconstrained decoding can lead to label drift, producing labels from unrelated tasks or the pretraining distribution. To mitigate this, we reformulate task-agnostic inference to operate on a refined input space constructed using three components:

**1) Representative Input Sampling.** To construct a compact yet informative training subset, we select $k$ representative samples per class via clustering, improving upon the random sampling strategy used in prior work (Razdaibiedina et al., 2023). The data set $\mathcal{D} = \{(x_j, y_j)\}$ is partitioned by label $y \in \mathcal{Y}$, and the sentences $\mathbf{e}_j = f_{\text{embed}}(x_j)$ are calculated using a pre-trained embedding model (`all-MiniLM-L6-v2`; (Reimers & Gurevych, 2019)). K-Means clustering is performed within each class to ensure intra-class diversity, and the top $k/C$ samples closest to each cluster center are selected based on cosine similarity: $\text{sim}(\mathbf{e}, \mathbf{c}) = \frac{\mathbf{e} \cdot \mathbf{c}}{\|\mathbf{e}\|\|\mathbf{c}\|}$. This yields a balanced subset $\mathcal{D}_{\text{rep}}$ that spans the semantic space of each class.

**2) Task Identification.** In many datasets, task categories are not explicitly provided, and labels often appear in non-descriptive formats (e.g., {0,1} or {choice1, choice2}) that are ambiguous without task context. To resolve this, we implement a hierarchical task identification module that infers the task type $t^*$ given a candidate label set $\mathcal{Y}_i$ and a sample input $x$. The process begins with

*rule-based heuristics*, which match label tokens and input structure to predefined task templates (e.g., "positive/negative" for sentiment analysis, or premise-hypothesis pairs with "entailment/contradiction" for NLI). If this fails, we fallback to *zero-shot task classification* using a lightweight generative language model (e.g., Phi-3.5). The model is prompted to infer the task type and remap non-descriptive labels into meaningful textual tokens: $(t^*, \tilde{\mathcal{Y}}_i) = \textbf{LLM}(x, \mathcal{Y}_i)$. This remapping enables consistent semantic interpretation across tasks, making it possible to apply decoding constraints in the task-agnostic setting.

**3) Constrained Decoding.** With remapped label set $\mathcal{L}_i = \{\ell_1, \ldots, \ell_K\}$, we apply constrained decoding at inference. At each decoding step $t$, we restrict the softmax to only allow tokens from $\mathcal{L}_i$:

$$\tilde{P}(y_t \mid y_{<t}, x) = \text{softmax}(M \odot \mathbf{z}_t), \quad M_j = \mathbf{1}[v_j \in \mathcal{L}_i],$$

where $\mathbf{z}_t$ are the raw logits and $M$ is a binary mask over the vocabulary $\mathcal{V}$.

## 3.2 Prompt Pool Compression via Gradient-Guided Selection

Prompt-based CL methods typically allocate one prompt per task, causing the prompt pool to grow as $\mathcal{O}(N)$. To achieve scalable long-horizon continual learning, we propose a dynamic selection–compression strategy based on gradient relevance. Let $\mathcal{P} = \{\mathbf{p}_1, \ldots, \mathbf{p}_{t-1}\}$ be the prompt pool before task $T_t$. For each $\mathbf{p}_j \in \mathcal{P}$, we compute the average gradient norm over new task data $D_t$:

$$g_j = \frac{1}{|D_t|} \sum_{(x,y) \in D_t} \|\nabla_{\mathbf{p}_j} L_t(f(x; \mathbf{p}_j), y)\|_2. \tag{1}$$

A large $g_j$ indicates that the task substantially updates $\mathbf{p}_j$, suggesting distinct knowledge worth preserving. Conversely, small $g_j$ values imply redundancy with the current task. Prompts are partitioned using the threshold

$$\tau = \mu_g + \alpha \sigma_g, \tag{2}$$

where $\mu_g, \sigma_g$ are the mean and standard deviation of $\{g_j\}$, and $\alpha$ is a tunable hyperparameter:

$$\mathcal{P}_{high} = \{\mathbf{p}_j : g_j > \tau\}, \quad \mathcal{P}_{low} = \{\mathbf{p}_j : g_j < \tau\}. \tag{3}$$

Although equation 1 only measures prompt–task interaction, we observed that low-gradient prompts are often highly redundant (average cosine similarity $\geq 0.87$, Euclidean radius $R < 0.45$). When similarity is high, discarding them has little effect; but when $\mathcal{P}_{low}$ is diverse, removal risks losing transferable information. To mitigate this, we aggregate $\mathcal{P}_{low}$ into a single embedding $\mathbf{p}_{agg}$ using gradient-weighted averaging:

$$\mathbf{p}_{agg} = \sum_{\mathbf{p}_j \in \mathcal{P}_{low}} w_j \mathbf{p}_j, \quad w_j = \frac{g_j}{\sum_{\mathbf{p}_k \in \mathcal{P}_{low}} g_k}. \tag{4}$$

This ensures that relatively more informative low-gradient prompts contribute proportionally. The updated pool for task $T_t$ becomes $\mathcal{P}' = \mathcal{P}_{high} \cup \{\mathbf{p}_{agg}\}$.

This mechanism preserves critical knowledge while substantially reducing memory and inference costs. *Notably, experiments show minimal degradation under compression, confirming that low-gradient prompts contribute little to future tasks.*

**Why reduce storage?** For every new task, all existing prompts are re-evaluated: only those with high gradient contributions are kept, while low-gradient or redundant ones are merged into a single aggregated prompt. This continual compression ensures that the pool size shrinks whenever redundancy is found. For example, after completing 15 tasks, ProgPrompt accumulates 15 prompts, whereas GRID retains 5, reducing storage by about 66.7% (Table 6). A smaller pool also decreases the number of prompts involved in forward/backward passes, which helps lower training time and can partly balance the additional gradient cost.

## 4 Experiments

### 4.1 Datasets and Baselines

**Datasets.** Following Razdaibiedina et al. (2023), we evaluate in few-shot continual-learning "long-sequence" settings with six 15-task sequences (order L1-L6). L1-L3 are taken from prior work,

while L4-L6 are newly constructed to study task difficulty progressions (easy→hard, hard→easy, and mixed). In addition, we introduce a *Negative Transfer* benchmark (order NT1–NT3), each containing 9 tasks with deliberately dissimilar transitions to induce transfer degradation. Detailed construction details are provided in the appendix B.2. We use T5 and Flan-T5 models, following prior continual learning work (Wu et al., 2024; Qin & Joty, 2021; Zhu et al., 2022; Lester et al., 2021; Wang et al., 2023b; Liang et al., 2023), covering sizes from T5-small (60M) to T5-large (770M), and additionally T5-3B to demonstrate scalability of our approach.

**Baseline Methods for Comparison.** We evaluate the proposed method against a total of 7 baseline methods[1]: (1) **Finetune** (Wang et al., 2020), (2) **Prompt Tuning** (Lester et al., 2021; Qin & Joty, 2021), (3) **Data Replay** (de Masson D'Autume et al., 2019), (4) **LFPT5** (Qin & Joty, 2021), (5) **Per-task Prompt** (Lester et al., 2021; Qin & Joty, 2021), (6) **ProgPrompt** (Razdaibiedina et al., 2023), and (7) **SHLPT** (Wu et al., 2024). We report average test accuracy, backward transfer (BWT), and forward transfer (FWT), defined by Lopez-Paz & Ranzato (2017) with results averaged over three runs; detailed metric definitions and additional results are provided in Appendix B.

## 4.2 AVERAGE ACCURACY

Our method excels in the long-sequence experiments (L1-L3; Table 2), achieving an average accuracy of 79.2% and surpassing competitive baselines such as SHLPT, the most recent state-of-the-art method. These results validate the effectiveness of our gradient-driven pruning approach in reducing redundancy while preserving essential task knowledge. Detailed results across T5 and Flan-T5 variants (Table 13 in the Appendix) show consistent gains, particularly on larger models (e.g., +3.2% for T5-large, +5.2% for FT5-base). Under the Negative

Table 2: Average test-set accuracy on long-sequence order L1–L6 with T5-large. The **DR** column indicates whether the method uses data replay (✓) or not (✗).

| Method | L1 | L2 | L3 | L4 | L5 | L6 | DR | Avg |
|--------|------|------|------|------|------|------|-----|------|
| Finetune | 8.3 | 8.7 | 7.8 | 7.9 | 8.1 | 8.9 | ✗ | 8.3 |
| Prompt Tuning | 8.8 | 9.5 | 8.1 | 9.3 | 9.4 | 9.4 | ✗ | 9.1 |
| Data Replay | 56.2 | 54.3 | 53.5 | 54.8 | 54.2 | 55.3 | ✓ | 54.7 |
| LFPT5 | 70.8 | 69.2 | 69.4 | 68.2 | 69.4 | 68.5 | ✓ | 69.3 |
| Per-task Prompt | 75.0 | 75.6 | 76.2 | 74.8 | 75.9 | 73.6 | ✗ | 75.2 |
| ProgPrompt | 75.7 | 78.6 | 74.3 | 75.05 | 77.10 | 75.46 | ✗ | 76.0 |
| SHLPT | 77.4 | 77.9 | 78.8 | 78.4 | 78.2 | **76.2** | ✗ | 77.8 |
| GRID* | **79.1** | **80.7** | **81.1** | **79.0** | **79.8** | 75.5 | ✗ | **79.2** |

Transfer benchmarks (Order NT1–NT3; Table 16), we also observe clear improvements (e.g., +3.8% on FT5-base, +3.6% on FT5-large), highlighting its robustness even in low-task-similarity scenarios.

## 4.3 BACKWARD TRANSFER ANALYSIS

Table 3 and Table 11 (Appendix) report BWT scores across long-sequence and model variants. GRID consistently achieves substantially less negative BWT than ProgPrompt and SHLPT, cutting forgetting by nearly half on average. The gains hold across all T5 and Flan-T5 model sizes, with relative improvements exceeding 50%. Interestingly, while larger models achieve higher accuracy (Table 13), they suffer greater forgetting, whereas smaller models retain prior knowledge better, likely due to less aggressive adaptation to new tasks. To visualize BWT dynamics, Figure 2 displays heatmaps of BWT scores for Order L1. Compared to the progprompt (A) and SHLPT (B), our method (C) exhibits brighter regions in the lower triangle, indicating stronger retention of earlier tasks. The difference heatmap (D,E) highlights

Table 3: BWT Score on T5-large across order (L1-L6) against the stronger baselines ProgPrompt and SHLPT.. Less negative values indicate reduced forgetting.

| Order | ProgPrompt | SHLPT | GRID |
|-------|-----------|---------|------|
| L1 | -0.7275 | -0.6123 | **-0.3243** |
| L2 | -0.7625 | -0.6870 | **-0.3336** |
| L3 | -0.6137 | -0.5174 | **-0.3979** |
| L4 | -0.6257 | -0.5042 | **-0.3912** |
| L5 | -0.6351 | -0.4187 | **-0.2956** |
| L6 | -0.6416 | -0.5840 | **-0.3512** |
| Avg | -0.6677 | -0.5539 | **-0.3490** |

widespread positive gains (blue), especially for early tasks, underscoring the effectiveness of our approach. Figure 3 shows per-task BWT differences between our method and the baseline. Positive values indicate improved retention, while negative values signify degradation. Notably, our method yields significant BWT improvements on challenging tasks like `boolq`, `sst2`, and `dbpedia_14`, while maintaining parity or improvement on most others. Additional analysis of task order effects,

---

[1]We were unable to compare with Q-Tuning (Guo et al., 2024), since it follows a different setup and its codebase is not publicly available for reproduction in our setting due to its company privacy restrictions.

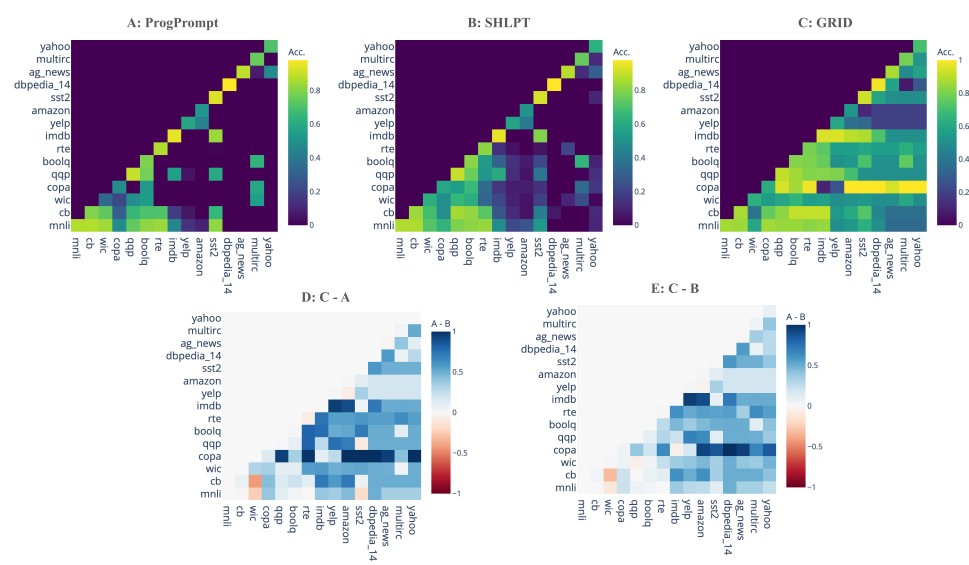

Figure 2: Heatmaps of backward transfer scores on previous tasks for Order L1. (A) Progressive Prompts, (B) SHLPT, (C) GRID, and differences (D) C–A, (E) C–B.

including comparisons between Order L4, L5, and L6, is provided in Section C.1, Tables 13, 14 and 17.

## 4.4 FORGOTTEN TASK COUNT

We define the Forgotten Task Count (FTC) as the number of tasks whose accuracy falls below a threshold relative to their standalone performance. Formally, task $T_i$ is forgotten at step $t$ if $a_i^{(t)} < \tau \cdot \min_j a_j^{(0)}$, where $a_i^{(0)}$ is the accuracy of $T_i$ when trained alone and $\tau \in (0, 1)$. Using 0 as a cutoff is misleading since accuracies may degrade to small but nonzero values; our formulation instead provides a principled absolute threshold.[2]. GRID reduces forgetting dramatically compared to both ProgPrompt and SHLPT. On average, GRID forgets only 13.8 tasks, compared to 78.7 for ProgPrompt and 64.0 for SHLPT. These results highlight that GRID mitigates catastrophic forgetting in long-horizon continual learning, while preserving task-relevant knowledge in a compact form. For results across different model variants, please refer to Table 14 in the Appendix.

Table 4: Comparison of forgotten task counts across various task order L1-L6.

| Order | ProgPrompt | SHLPT | GRID |
|-------|-----------|-------|------|
| L1 | 77 | 64 | **11** |
| L2 | 72 | 58 | **5** |
| L3 | 80 | 69 | **18** |
| L4 | 87 | 73 | **26** |
| L5 | 71 | 58 | **10** |
| L6 | 85 | 62 | **13** |
| **Avg** | 78.7 | 64.0 | **13.8** |

## 4.5 SCALABILITY TO LARGER MODELS

We have also run experiments on a larger model, T5-3B, to validate scalability. The results (Table 5) show that GRID maintains its advantages at this larger scale. These findings suggest that GRID's core mechanisms scale effectively to models with billions of parameters.

## 4.6 ABLATION STUDY

Table 5: Comparison on T5-3B model

| Order | Method | Accuracy | BWT | Forgot. |
|-------|--------|----------|-----|---------|
| L1 | ProgPrompt | 74.86 | -0.7161 | 69 |
| | SHLPT | 77.18 | -0.5841 | 57 |
| | GRID | **78.54** | **-0.4445** | **21** |
| L2 | ProgPrompt | 75.08 | -0.7207 | 69 |
| | SHLPT | 79.29 | -0.6343 | 51 |
| | GRID | **81.94** | **-0.5255** | **17** |
| L3 | ProgPrompt | 74.14 | -0.5840 | 81 |
| | SHLPT | 79.56 | -0.5632 | 61 |
| | GRID | **81.85** | **-0.4152** | **33** |

To evaluate the contribution of each component in the GRID framework, we conduct an ablation study on task order L1, L2, and L3, measuring BWT under four variants: (0) the full **G.R.I.D.** model,

[2]In our task list, Amazon had the lowest standalone accuracy ($\sim$0.50); setting $\tau = 0.4$ yields an absolute threshold of 0.20.

(1) without gradient-based prompt selection (G), (2) without constrained decoding (D), (3) without representative input selection (R), and (4) without all components. As shown in Table 6, the full model yields the highest BWT for both T5-large (**-0.3519**) and FlanT5-large (**-0.3492**). Excluding gradient-based selection has minimal effect on BWT, consistent with its role in reducing memory rather than directly improving retention: GRID requires only **200 KB** of storage versus **600 KB** for ProgPrompt, a **66.7% reduction**. In contrast, removing constrained decoding causes a pronounced BWT drop, and eliminating both components leads to the worst performance (**-0.7012** for T5-large, **-0.7106** for FlanT5-large). Overall, these results underscore constrained decoding as the primary driver of backward transfer, with gradient-based selection offering complementary scalability.

Table 6: Ablation on GRID over orders L1–L3. We report BWT (less negative is better), average across orders, prompt memory size in KB (slots), and GPU time per run on A100 (40GB) with batch size 8.

| Model | Variant | L1 | L2 | L3 | Avg | Memory | GPU (h:m) |
|---|---|---|---|---|---|---|---|
| | **(0) G.R.I.D.** | **−0.3243** | **−0.3310** | **−0.3979** | **−0.3511** | **200** | 27:08 |
| | (1) w/o G | −0.3254 | −0.3321 | −0.3895 | −0.3490 | 600 | 25:35 |
| T5-large | (2) w/o G,D | −0.7032 | −0.7589 | −0.5967 | −0.6863 | 600 | 26:42 |
| | (3) w/o G,D,R | −0.7155 | −0.7612 | −0.5993 | −0.6954 | 600 | 24:58 |
| | (4) w/o all | −0.7275 | −0.7625 | −0.6137 | −0.7012 | 600 | 23:58 |
| | **(0) G.R.I.D.** | **−0.3115** | **−0.3456** | **−0.3705** | **−0.3425** | **200** | 27:10 |
| | (1) w/o G | −0.3145 | −0.3510 | −0.3723 | −0.3459 | 600 | 25:38 |
| FT5-large | (2) w/o G,D | −0.7321 | −0.7428 | −0.6056 | −0.6935 | 600 | 26:37 |
| | (3) w/o G,D,R | −0.7355 | −0.7499 | −0.6176 | −0.7014 | 600 | 23:46 |
| | (4) w/o all | −0.7679 | −0.7444 | −0.6195 | −0.7106 | 600 | 23:47 |

**Runtime and Hardware.** In addition to the A100 experiments in Table 6, we evaluated GRID on H100 GPUs (80GB) with batch size 16. On H100, long task sequences averaged $\sim 11$ hours per run with peak memory usage of $\sim 65$ GB, compared to $\sim 27$ hours and $\sim 21$ GB on A100 (40GB, batch size 8). These results demonstrate that GRID scales efficiently across hardware generations.

**Role of Representative Inputs.** We empirically find that increasing the number of representative samples beyond 1k per class leads to accuracy saturation with only negligible improvements. This indicates that larger sample sizes may not yield substantial performance gains (Table 7). By contrast, clustering provides both data efficiency and better generalization, highlighting the importance of representative input selection in GRID.

**Impact of Prompt Selection Strategies.** As shown in Table 8, our gradient-based method consistently yields the fewest forgotten tasks and achieves competitive or superior BWT across all settings. FIFO occasionally matches our BWT but retains more redundant prompts, resulting in higher forgetting. Random selection performs worst, with unstable BWT and the largest number of forgotten tasks.

Table 7: Accuracy for different datasets with varying sample sizes (k) highlights how model performance scales with more training samples.

| Dataset | k | Acc. |
|---|---|---|
| | 20 | 0.5203 |
| | 200 | 0.9674 |
| DBPedia | 1000 | 0.9880 |
| | 2000 | 0.9892 |
| | 20 | 0.0000 |
| | 200 | 0.0000 |
| Amazon | 1000 | 0.5136 |
| | 2000 | 0.5534 |
| | 20 | 0.0000 |
| AG News | 200 | 0.8180 |
| | 1000 | 0.8900 |

Table 8: Comparison of prompt selection strategies (Random, FIFO, Gradient-based) for T5-large and FlanT5-large across order L1–L3. Metrics: average accuracy (Acc), BWT, and number of forgotten tasks (FT).

| Model | Strategy | L1 | | | L2 | | | L3 | | |
|---|---|---|---|---|---|---|---|---|---|---|
| | | Acc | BWT | FT | Acc | BWT | FT | Acc | BWT | FT |
| | Random | 76.81 | -0.3482 | 15 | **82.07** | -0.3549 | 7 | **82.06** | -0.4065 | 21 |
| T5-large | FIFO | 76.08 | -0.3389 | 14 | 57.39 | -0.3780 | 11 | 81.34 | -0.3962 | 20 |
| | Ours | **79.12** | **-0.3243** | **11** | 80.69 | **-0.3336** | **5** | 81.09 | **-0.3979** | **18** |
| | Random | 78.75 | -0.3767 | 18 | 76.15 | -0.3835 | 12 | 76.28 | -0.3705 | 25 |
| FlanT5-large | FIFO | 76.62 | -0.3481 | 16 | 77.05 | -0.3981 | 16 | 76.60 | -0.3633 | 24 |
| | Ours | **79.77** | **-0.3115** | **15** | **78.25** | **-0.3656** | **10** | 76.09 | **-0.3544** | **24** |

## 5 RELATED WORK

**Continual Learning.** Continual learning (CL) involves learning from a sequence of tasks without full access to previous task data, aiming to preserve prior knowledge and enable positive transfer. A key challenge is catastrophic forgetting (McCloskey & Cohen, 1989), where updates to model parameters on new data erode earlier knowledge. CL strategies are categorized into three approaches: memory-based methods store and replay past task data to mitigate forgetting (Shin et al., 2017; Bang et al., 2021); regularization-based methods penalize deviations from important parameters to retain knowledge without stored data (Kirkpatrick et al., 2017; Zenke et al., 2017). Du et al. (2024) adopts a gradient-masking strategy by updating only high-activation model parameters, achieving task-agnostic and rehearsal-free CL; and architecture-based methods expand the model by adding new components for each task (Rusu et al., 2016; Yoon et al., 2018). These methods face scalability issues in large pretrained models, motivating the development of parameter-efficient CL techniques using lightweight components like prompts and adapters (Xu et al., 2023; Rücklé et al., 2021).

**Continual Prompt Tuning.** Prompt tuning (Lester et al., 2021; Li & Liang, 2021; Gu et al., 2021; Wang et al., 2023b; Jia et al., 2022) adapts large language models (LLMs) by learning a small set of continuous vectors, or soft prompts, prepended to the input tokens. Unlike full finetuning, it updates only the prompts while freezing model parameters, achieving competitive or superior performance with lower computational and memory cost. Continual prompt tuning (CPT) (Zhu et al., 2022; Yin et al., 2022; Ermis et al., 2022; Wang et al., 2022a) extends this idea to the continual learning (CL) setting, where models adapt to evolving task sequences. A substantial body of work has focused on improving CPT's ability to retain and transfer knowledge, using techniques such as prompt concatenation (Razdaibiedina et al., 2023), parameter sharing (Wang et al., 2022b), and weighted prompt summation (Jiang et al., 2023). However, existing approaches often rely on large memory buffers to mitigate forgetting (Zhu et al., 2022; Ermis et al., 2022), impractical in privacy- or resource-constrained scenarios. Moreover, task-specific prompts suffer from latent forgetting under task-agnostic inference (Guo et al., 2024), while methods that expand prompt pools over tasks (Razdaibiedina et al., 2023; Wang et al., 2022a; Smith et al., 2023) face scalability and efficiency bottlenecks as tasks accumulate.

**Gradient-Based Data and Parameter Selection.** Gradient-based strategies have been extensively studied in data selection and coreset construction (Aljundi et al., 2019b), where the goal is to identify a representative or influential subset of training data (Borovicka et al., 2012; Rolf et al., 2021). Methods such as Coreset Selection (Killamsetty et al., 2021; Mirzasoleiman et al., 2020; Hao et al., 2023) and Gradient Matching (Aljundi et al., 2019b) leverage the similarity or norm of training gradients to preserve performance while reducing dataset sizes. In the context of prompt learning, some recent works have begun to reduce the prompt memory. For example, Q-Tuning (Guo et al., 2024) uses a PCA-based eviction and L2P (Wang et al., 2022b) keeps updating the same set of prompts over time. In contrast, our work draws inspiration from the coreset literature and extends it to the prompt space by dynamically evaluating gradient norms of prompt embeddings, allowing us to merge less informative prompts and improve scalability under long task sequences.

## 6 CONCLUSION

In this work, we introduce GRID, a unified framework that renders prompt-based continual learning both scalable and resilient to forgetting. Unlike prior approaches that depend on task-aware inference or accumulate ever-growing prompt pools, GRID tackles two key challenges: latent forgetting under task-agnostic inference and the inefficiency of unbounded prompt memory. It integrates constrained decoding with gradient-guided prompt selection and compression, enabling consistent label generation, compact memory usage, and enhanced knowledge retention. Our method substantially improves backward transfer, reducing forgotten tasks by a large margin, while maintaining competitive forward transfer.

Future work could extend GRID to also enable positive backward knowledge transfer, allowing new tasks to refine earlier prompts and further boost their performance. Additional directions include scaling to much longer task streams, applying the framework to other types of foundation models.

## 7 REPRODUCIBILITY STATEMENT

We have made significant efforts to ensure the reproducibility of our results. Our supplementary code submission contains a cleaned and anonymized implementation of GRID. The repository includes core training and evaluation scripts, as well as representative training and test datasets for quality checks. Detailed experimental settings and hyperparameters are provided in the main paper and appendix.

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

# SUPPLEMENTARY MATERIALS

## A ALGORITHMIC DETAILS

**Algorithm 1** Representative Sample Selection

**Require:** Dataset $\mathcal{D} = \{(x_j, y_j)\}$, samples per class $k$, embedding model $f_{embed}$, number of clusters $C$
**Ensure:** Subset $\mathcal{D}_{rep}$
 1: Initialize $\mathcal{D}_{rep} \leftarrow \emptyset$
 2: **for** each label $y$ **do**
 3: $\quad \mathcal{D}_y \leftarrow \{x_j \mid y_j = y\}$
 4: $\quad$ Embed each $x_j$: $\mathbf{e}_j \leftarrow f_{embed}(x_j)$
 5: $\quad$ Run K-Means on $\{\mathbf{e}_j\}$ into $C$ clusters
 6: $\quad$ **for** each cluster $c$ **do**
 7: $\qquad$ Select top $k/C$ samples by similarity to center
 8: $\qquad$ Add to $\mathcal{D}_{rep}$
 9: $\quad$ **end for**
10: $\quad$ **if** selected $< k$ **then**
11: $\qquad$ Add random samples to reach $k$ total
12: $\quad$ **end if**
13: **end for**
14: Shuffle and return $\mathcal{D}_{rep}$

**Algorithm 2** Prompt Selection and Aggregation

**Require:** Prompt pool $\mathcal{P} = \{\mathbf{p}_j\}$, new task data $D_t$, thresholds $\alpha, \beta$
**Ensure:** Updated prompt pool $\mathcal{P}'$
 1: **for** each $\mathbf{p}_j \in \mathcal{P}$ **do**
 2: $\quad$ Compute gradient norm $g_j$ via Eq. 1
 3: **end for**
 4: Compute $\mu_g, \sigma_g$; threshold $\tau$ via Eq. 2
 5: Classify prompts into $\mathcal{P}_{high}$ and $\mathcal{P}_{low}$ via Eq. 3
 6: **if** $\mathcal{P}_{low} \neq \emptyset$ **then**
 7: $\quad$ Compute aggregation $\mathbf{p}_{agg}$ via Eq. 4
 8: $\quad \mathcal{P}' \leftarrow \mathcal{P}_{high} \cup \{\mathbf{p}_{agg}\}$
 9: **else**
10: $\quad \mathcal{P}' \leftarrow \mathcal{P}_{high}$
11: **end if**
12: Append new prompt $\mathbf{p}_t$ to $\mathcal{P}'$
13: **return** $\mathcal{P}'$

## B FURTHER IMPLEMENTATION DETAILS

### B.1 DATASETS

Table 9: Overview of the 15 datasets used in our CL experiments, including their evaluation metrics. Datasets from CL benchmark (Zhang et al., 2015), GLUE (Wang et al., 2018), and SuperGLUE (Wang et al., 2019) were utilized, along with the IMDB movie reviews dataset. For tasks with two evaluation metrics, we report the average as the final performance measure.

| Dataset name | Category | Task | Domain | Metric |
|---|---|---|---|---|
| 1. Yelp | CL benchmark | sentiment analysis | Yelp reviews | accuracy |
| 2. Amazon | CL benchmark | sentiment analysis | Amazon reviews | accuracy |
| 3. DBpedia | CL benchmark | topic classification | Wikipedia | accuracy |
| 4. Yahoo | CL benchmark | QA | Yahoo Q&A | accuracy |
| 5. AG News | CL benchmark | topic classification | news | accuracy |
| 6. MNLI | GLUE | NLI | various | accuracy |
| 7. QQP | GLUE | paraphrase detection | Quora | accuracy & F1 |
| 8. RTE | GLUE | NLI | news, Wikipedia | accuracy |
| 9. SST2 | GLUE | sentiment analysis | movie reviews | accuracy |
| 10. WiC | SuperGLUE | word sense disambiguation | lexical databases | accuracy |
| 11. CB | SuperGLUE | NLI | various | accuracy |
| 12. COPA | SuperGLUE | QA | blogs, encyclopedia | accuracy |
| 13. BoolQ | SuperGLUE | boolean QA | Wikipedia | accuracy |
| 14. MultiRC | SuperGLUE | QA | various | F1 & EM |
| 15. IMDB | Other | sentiment analysis | movie reviews | accuracy |

### B.2 TASK SEQUENCE ORDERS

**Long-Sequence CL Setting:** Following the approach of (Razdaibiedina et al., 2023), we consider a total of 15 distinct tasks. These consist of five datasets from the CL benchmark (Zhang et al., 2015): AG News (topic classification), Amazon Reviews (sentiment analysis), Yelp Reviews (sentiment analysis), DBpedia (Wikipedia text classification), Yahoo Answers (Q&A classification), four tasks

Table 10: Task sequence orders used in our CL experiments. Order L1-L3 correspond to long-sequence CL benchmarks. Order L4-L6 are our proposed custom sequences, with L4 representing an easy-to-hard order, L5 representing a hard-to-easy order, and L6 representing a mixed order. NT1, NT2, and NT3 represent different task evaluation orders designed to examine negative transfer across various task sequences.

| Order | Task Sequence |
|---|---|
| L1 | mnli → cb → wic → copa → qqp → boolq → rte → imdb → yelp → amazon → sst2 → dpedia → ag → multirc → yahoo |
| L2 | multirc → boolq → wic → mnli → cb → copa → qqp → rte → imdb → sst2 → dpedia → ag → yelp → amazon → yahoo |
| L3 | yelp → amazon → mnli → cb → copa → qqp → rte → imdb → sst2 → dpedia → ag → yahoo → multirc → boolq → wic |
| L4 | sst2 → imdb → yelp → amazon → ag → yahoo → dbpedia → mnli → rte → cb → qqp → copa → boolq → wic → multirc |
| L5 | multirc → wic → boolq → copa → qqp → cb → rte → mnli → dbpedia → yahoo → ag → amazon → yelp → imdb → sst2 |
| L6 | sst2 → copa → ag → imdb → mnli → yahoo → rte → yelp → qqp → cb → amazon → dbpedia → boolq → wic → multirc |
| NT1 | multirc → wic → mnli → cb → rte → qqp → yahoo → yelp → amazon |
| NT2 | amazon → yelp → yahoo → qqp → rte → cb → mnli → wic → multirc |
| NT3 | yahoo → mnli → amazon → cb → yelp → rte → qqp → multirc → wic |

from the GLUE benchmark (MNLI, QQP, RTE, SST2) (Wang et al., 2018), five tasks from the SuperGLUE benchmark (WiC, CB, COPA, MultiRC, BoolQ) (Wang et al., 2019), and the IMDB movie reviews dataset for sentiment analysis (Maas et al., 2011). We evaluate our methods across six different task sequence orders. Three of these sequences follow standard long-sequence continual learning orders that have been used in prior works (e.g., L1–L3) (Razdaibiedina et al., 2023; Guo et al., 2024). We additionally propose three novel task orders: (1) *Order L4: easy-to-hard*, where tasks are grouped by category and simpler tasks such as binary sentiment classification are introduced first, followed by more complex reasoning tasks like MultiRC and COPA; (2) *Order L5: hard-to-easy*, which reverses this progression; and (3) *Order L6: mixed*, where tasks from different categories and difficulty levels are randomly interleaved. These variations allow us to assess how task presentation order and semantic similarity affect forgetting, transfer, and model generalization in long-horizon continual learning scenarios.

**Negative Transfer Benchmark.** To evaluate the robustness of our approach against negative transfer (Pan, 2010), we introduce a *Negative Transfer Benchmark* (NT1, NT2, NT3) inspired by the SHLPT framework (Wu et al., 2024). Specifically, we identify task pairs that exhibit negative transfer based on the analysis provided in the SHLPT. Using these insights, we create three task sequences that are likely to cause negative transfer, meaning earlier tasks make it harder to learn later ones. This helps us test how well our method handles such challenges.

## B.3 IMPLEMENTATION DETAILS

In our experiments, we focus on encoder-decoder transformer-based architecture, adopting both the original T5(Raffel et al., 2020) and Flan-T5(Chung et al., 2022) as our backbone architectures. To study the impact of model size, we perform evaluations using the `small`, `base`, `large`, and `3B` variants. This setup allows us to assess the scalability of our method across a range of model capacities.

## B.4 EXPERIMENT DETAILS

In all our experiments with T5 and Flan-T5 backbones, we fix the prompt length to 10 tokens, train for 10 epochs, and use 1k representative samples per class.[3] This configuration is applied uniformly

---

[3]We could not experiment with 300 epochs and 20 samples per class (as in ProgPrompt (Razdaibiedina et al., 2023)) due to resource constraints.

Table 11: Backward Transfer (BWT) comparison between our method and ProgPrompt across multiple task orderings (Order L1-L6) for T5 and Flan-T5 models. Less negative scores indicate reduced forgetting. Our method consistently improves BWT across all model variants.

| Model | Setting | Order 4 | Order 5 | Order 6 | Order A | Order B | Order C | Avg | Imp. (%) |
|-------|---------|---------|---------|---------|---------|---------|---------|-----|----------|
| T5-small | ProgPrompt | -0.6081 | -0.5414 | -0.5126 | -0.5243 | -0.5377 | -0.5006 | -0.5374 | **54.4** ↑ |
| | Ours | -0.1402 | -0.1894 | -0.2504 | -0.3141 | -0.2734 | -0.3006 | **-0.2447** | |
| FT5-small | ProgPrompt | -0.5711 | -0.5574 | -0.4916 | -0.4914 | -0.5075 | -0.4987 | -0.5196 | **68.4** ↑ |
| | Ours | -0.0518 | -0.1449 | -0.2106 | -0.2129 | -0.1517 | -0.2396 | **-0.1686** | |
| T5-base | ProgPrompt | -0.6918 | -0.6520 | -0.5948 | -0.5956 | -0.6471 | -0.5881 | -0.6282 | **52.7** ↑ |
| | Ours | -0.3060 | -0.2404 | -0.3549 | -0.2893 | -0.3439 | -0.3259 | **-0.3101** | |
| FT5-base | ProgPrompt | -0.6092 | -0.7222 | -0.6488 | -0.6273 | -0.6631 | -0.6225 | -0.6489 | **58.7** ↑ |
| | Ours | -0.2616 | -0.3243 | -0.3319 | -0.3245 | -0.3240 | -0.2743 | **-0.3067** | |
| T5-large | ProgPrompt | -0.7275 | -0.7625 | -0.6137 | -0.6257 | -0.6351 | -0.6416 | -0.6677 | **51.7** ↑ |
| | Ours | -0.3243 | -0.3336 | -0.3979 | -0.3912 | -0.2956 | -0.3512 | **-0.3490** | |
| FT5-large | ProgPrompt | -0.7679 | -0.7444 | -0.6195 | -0.6616 | -0.7107 | -0.6631 | -0.6945 | **50.4** ↑ |
| | Ours | -0.3115 | -0.3656 | -0.3544 | -0.3514 | -0.3282 | -0.3540 | **-0.3442** | |

across the `small`, `base`, and `large` variants of both model families, ensuring a fair yet efficient comparison to prior work such as ProgPrompt (Razdaibiedina et al., 2023).

## C  ADDITIONAL RESULTS

### C.1  ORDER AND BWT ANALYSIS

One critical observation from Table 11 and Table 14 is that Order L5 consistently outperforms Order L4 in both BWT scores and the number of forgotten tasks. Specifically, for the same set of tasks, Order L5 shows less negative BWT scores, indicating reduced forgetting. Additionally, the number of forgotten tasks is lower in Order L5, further suggesting that harder tasks at the beginning of training help the model retain earlier knowledge more effectively. This observation highlights the benefit of starting with more difficult tasks, as it leads to better long-term retention and improved overall performance in continual learning. One also notes that Order L6 performs better than Order L4 but slightly less effectively than Order L5, indicating that a mixed task sequence provides a balanced approach to task progression, offering a compromise between the benefits of easy-to-hard and hard-to-easy learning.

**Forgotten Task Count.** We further analyze the number of forgotten tasks across various settings in Table 14. Our method significantly reduces the number of forgotten tasks compared to the Progressive Prompts baseline. For instance, in the case of `T5-small`, the number of forgotten tasks drops from as high as 93 in some orders to as low as 7 under our method. Similar improvements are observed for `T5-base`, where the count decreases from 90 to just 6 in the best case, and for `T5-large`, where it reduces from 87 to only 5. These improvements are consistent across all task orderings, including structured sequences such as Order L4 (easy-to-hard), Order L5 (hard-to-easy), and Order L6 (mixed).

## D  LIMITATIONS AND FUTURE WORK

While GRID demonstrates strong improvements in backward transfer and scalability, it has two primary limitations. First, our evaluation is limited to sequences of 15 tasks. To fully assess the scalability of our method, future work should test it on significantly longer task streams. Second, the current GRID framework, specifically the constrained decoding component is designed for encoder-decoder architectures. Although our gradient-based prompt selection strategy is model-agnostic and can be applied broadly, the full framework has not yet been extended to other foundational models. Exploring how to adapt GRID's decoding mechanism to these architectures is a promising direction for future research.

Table 12: Impact of prompt length on T5-large performance across Order L1-L3. We observe a trade-off: longer prompts improve BWT (less forgetting) but increase the number of forgotten tasks and can degrade accuracy. Optimal performance is generally observed at a moderate prompt length (e.g., 10).

| Prompt Length | L1 | | | L2 | | | L3 | | |
|---|---|---|---|---|---|---|---|---|---|
| | Acc | BWT | FT | Acc | BWT | FT | Acc | BWT | FT |
| 5 | 78.23 | -0.3167 | 10 | 81.73 | -0.3565 | 5 | 84.98 | -0.4286 | 15 |
| 10 | 79.12 | -0.3243 | 11 | 80.69 | -0.3336 | 5 | 81.09 | -0.3979 | 18 |
| 20 | 80.58 | -0.3396 | 12 | 79.37 | -0.3159 | 5 | 75.55 | -0.3206 | 24 |

Table 13: Average test accuracy across long-sequence task orderings (Order L1-L6) for T5 and Flan-T5 models. Our method consistently outperforms Progressive Prompts in nearly all configurations.

| Model | Setting | L1 | L2 | L3 | L4 | L5 | L6 | Avg |
|---|---|---|---|---|---|---|---|---|
| T5-small | ProgPrompt | 63.18 | 57.18 | 61.12 | 62.73 | 59.11 | 60.33 | 60.61 |
| | Ours | 59.75 | 64.97 | 59.90 | 64.64 | 64.51 | 62.39 | **62.69** |
| FT5-small | ProgPrompt | 60.32 | 59.25 | 59.43 | 59.18 | 59.35 | 59.96 | 59.58 |
| | Ours | 58.35 | 57.37 | 64.73 | 61.74 | 58.30 | 57.07 | 59.59 |
| T5-base | ProgPrompt | 71.34 | 67.60 | 69.57 | 69.71 | 72.43 | 69.93 | 70.10 |
| | Ours | 72.74 | 71.24 | 72.09 | 67.16 | 75.72 | 65.23 | 70.70 |
| FT5-base | ProgPrompt | 63.73 | 74.83 | 75.23 | 72.14 | 74.25 | 72.48 | 72.11 |
| | Ours | 77.24 | 78.08 | 77.16 | 78.32 | 73.53 | 65.92 | **75.04** |
| T5-large | ProgPrompt | 75.68 | 78.56 | 74.29 | 75.05 | 77.10 | 75.46 | 76.02 |
| | Ours | 79.12 | 80.69 | 81.09 | 79.05 | 79.79 | 75.54 | **79.21** |
| FT5-large | ProgPrompt | 79.81 | 77.51 | 74.30 | 79.13 | 79.09 | 79.56 | **78.23** |
| | Ours | 79.77 | 78.25 | 76.09 | 76.91 | 76.94 | 76.74 | 77.45 |

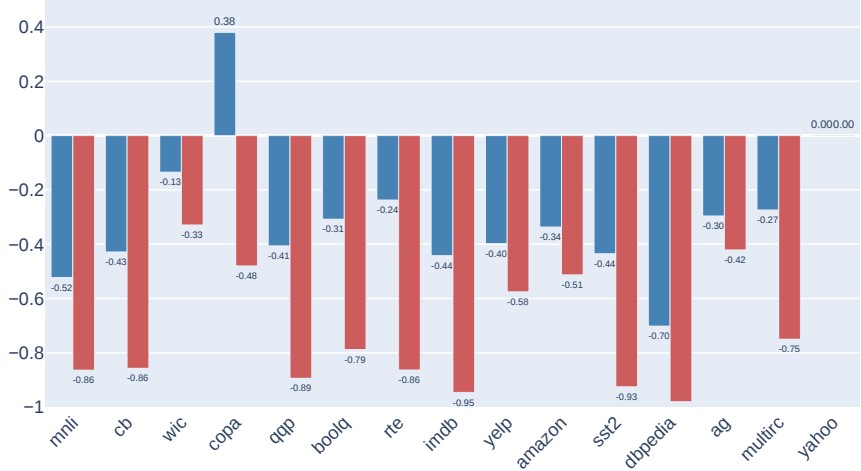

Figure 3: Per-task BWT comparison between our method (blue) and the baseline (red) for Order L1. Positive bars indicate improved retention of prior tasks. Our method shows significant BWT gains on several tasks (e.g., `copa`, `wic`, `yahoo`), demonstrating its effectiveness in mitigating forgetting across diverse task types.

Table 14: Comparison of forgotten task counts between Progressive Prompts (Baseline) and our method for different models across various task orders (order L1–L6).

| Model | Setting | L1 | L2 | L3 | L4 | L5 | L6 | Average | Improvement (%) |
|-------|---------|----|----|----|----|----|----|---------|-----------------|
| T5-small | ProgPrompt | 85 | 85 | 88 | 90 | 84 | 93 | 87.5 | 76.8% |
|          | Ours | **14** | **7** | **28** | **36** | **16** | **21** | **20.3** | |
| FT5-small | ProgPrompt | 93 | 95 | 95 | 94 | 97 | 93 | 94.5 | 76.7% |
|           | Ours | **17** | **14** | **28** | **26** | **20** | **27** | **22.0** | |
| T5-base | ProgPrompt | 88 | 82 | 76 | 90 | 83 | 88 | 84.5 | 80.9% |
|         | Ours | **15** | **6** | **19** | **29** | **13** | **15** | **16.2** | |
| FT5-base | ProgPrompt | 94 | 93 | 93 | 93 | 93 | 93 | 93.2 | 83.0% |
|          | Ours | **10** | **10** | **15** | **25** | **14** | **21** | **15.8** | |
| T5-large | ProgPrompt | 77 | 72 | 80 | 87 | 71 | 85 | 78.7 | 82.4% |
|          | Ours | **11** | **5** | **18** | **26** | **10** | **13** | **13.8** | |
| FT5-large | ProgPrompt | 92 | 93 | 93 | 93 | 91 | 93 | 92.5 | 79.8% |
|           | Ours | **15** | **10** | **24** | **24** | **14** | **25** | **18.7** | |

Table 15: Backward Transfer (BWT) scores on negative transfer benchmarks (NT1–NT3) comparing our method with Progressive Prompts across T5 and Flan-T5 models. Lower-magnitude (less negative) values indicate reduced forgetting. Our method consistently achieves better BWT, demonstrating robustness under task dissimilarity.

| Model | Setting | NT1 | NT2 | NT3 | Avg |
|-------|---------|-----|-----|-----|-----|
| T5-small | Baseline | -0.5783 | -0.4882 | -0.5133 | -0.5266 |
|          | Ours | -0.1957 | -0.1945 | -0.2043 | -0.1982 |
| FT5-small | Baseline | -0.4771 | -0.3924 | -0.4097 | -0.4264 |
|           | Ours | -0.0807 | -0.1601 | -0.1171 | -0.1193 |
| T5-base | Baseline | -0.5707 | -0.5834 | -0.5890 | -0.5810 |
|         | Ours | -0.1817 | -0.2103 | -0.2314 | -0.2078 |
| FT5-base | Baseline | -0.6719 | -0.5335 | -0.6116 | -0.6057 |
|          | Ours | -0.2911 | -0.2074 | -0.3162 | -0.2716 |
| T5-large | Baseline | -0.7088 | -0.6598 | -0.6536 | -0.6741 |
|          | Ours | -0.3154 | -0.3658 | -0.3077 | -0.3296 |
| FT5-large | Baseline | -0.7156 | -0.6628 | -0.6886 | -0.6890 |
|           | Ours | -0.2858 | -0.2888 | -0.3636 | -0.3127 |

Table 16: Average test accuracy on negative transfer task sequences (NT1–NT3) comparing our method with Progressive Prompts. Our method consistently outperforms the baseline across all models, demonstrating better generalization under reduced task similarity.

| Model | Setting | NT1 | NT2 | NT3 | Avg |
|-------|---------|-----|-----|-----|-----|
| T5-small | Baseline | 57.22 | 55.51 | 57.79 | 56.84 |
|          | Ours | 58.21 | 55.08 | 56.44 | 56.58 |
| FT5-small | Baseline | 51.57 | 46.02 | 48.36 | 48.65 |
|           | Ours | 50.40 | 50.54 | 49.77 | 50.24 |
| T5-base | Baseline | 61.18 | 65.75 | 66.42 | 64.45 |
|         | Ours | 63.34 | 63.46 | 63.30 | 63.37 |
| FT5-base | Baseline | 70.81 | 60.83 | 67.25 | 66.30 |
|          | Ours | 70.76 | 61.54 | 70.05 | 67.45 |
| T5-large | Baseline | 73.57 | 73.63 | 74.26 | 73.82 |
|          | Ours | 72.29 | 74.84 | 73.07 | 73.40 |
| FT5-large | Baseline | 75.98 | 73.98 | 74.84 | 74.93 |
|           | Ours | 76.28 | 71.56 | 74.72 | 74.19 |

Table 17: Forward Transfer (FWT) scores across various task orderings (Order L1-L6) for T5 and Flan-T5 models. Higher scores indicate better transfer to future tasks. Our method achieves comparable or improved FWT in most cases, demonstrating that reducing forgetting does not come at the cost of forward transfer.

| Model | Setting | L1 | L2 | L3 | L4 | L5 | L6 | Avg |
|---|---|---|---|---|---|---|---|---|
| T5-small | Baseline | -5.07 | -11.28 | -1.48 | -6.09 | -9.58 | -8.63 | -7.02 |
| | Ours | -5.82 | -0.11 | -5.55 | -0.96 | -1.11 | -3.33 | -2.81 |
| FT5-small | Baseline | -3.57 | -5.26 | -4.98 | -5.33 | -4.92 | -4.64 | -4.78 |
| | Ours | -2.53 | -4.09 | 3.33 | 0.25 | -3.02 | -4.64 | -1.78 |
| T5-base | Baseline | -3.02 | -7.02 | -4.80 | -4.55 | -1.58 | -4.23 | -4.20 |
| | Ours | -4.93 | -6.68 | -5.78 | -11.32 | -2.21 | -13.28 | -7.37 |
| FT5-base | Baseline | -13.91 | -2.08 | -1.48 | -4.90 | -2.60 | -4.53 | -4.92 |
| | Ours | 4.27 | 5.10 | 4.04 | 5.32 | 0.18 | -7.90 | 1.84 |
| T5-large | Baseline | -4.14 | -1.19 | -5.58 | -2.66 | -2.67 | -3.81 | -3.34 |
| | Ours | -2.82 | -1.20 | -0.90 | -3.11 | -2.20 | -6.28 | -2.75 |
| FT5-large | Baseline | -0.73 | -3.17 | -6.65 | -1.43 | -1.53 | -0.99 | -2.42 |
| | Ours | 2.35 | 0.75 | -1.48 | -0.64 | -0.60 | -0.90 | -0.09 |

Table 18: Comparison of FWT scores between Progressive Prompts (Baseline) and Ours across different models. The table reports average performance on different tasks order , including Order NT1, NT2, NT3, and the overall average (Avg).

| Model | Setting | NT1 | NT2 | NT3 | Avg |
|---|---|---|---|---|---|
| T5-small | Baseline | -5.86 | -7.62 | -5.62 | -6.37 |
| | Ours | -4.67 | -7.65 | -6.87 | -6.40 |
| FT5-small | Baseline | -3.93 | -9.99 | -7.88 | -7.27 |
| | Ours | -4.96 | -4.58 | -6.47 | -5.34 |
| T5-base | Baseline | -7.44 | -1.35 | -1.44 | -3.41 |
| | Ours | -6.52 | -5.56 | -6.64 | -6.24 |
| FT5-base | Baseline | -1.29 | -12.68 | -5.35 | -6.44 |
| | Ours | -1.43 | -11.72 | -2.05 | -5.07 |
| T5-large | Baseline | -2.55 | -2.34 | -1.62 | -2.17 |
| | Ours | -4.70 | -1.85 | -3.76 | -3.44 |
| FT5-large | Baseline | 0.09 | -2.19 | -0.76 | -0.95 |
| | Ours | 1.06 | -4.39 | -0.77 | -1.37 |

Table 19: Examples highlighting **inconsistent label mappings** in COPA under task-agnostic inference. Prior works like Progressive Prompts use manual conversions (e.g., 0 → "false"), often misaligning task semantics. GRID mitigates these inconsistencies by inferring appropriate labels automatically.

| Premise | Choice1 | Choice2 | Question | Label | ProgPrompt | GRID |
|---|---|---|---|---|---|---|
| The man lost the competition. | The competition was sabotaged. | He intimidated his competitors. | cause | 0 | false | choice1 |
| I regained composure from my fit of anger. | My heart pounded. | I took deep breaths. | cause | 1 | true | choice2 |
| The cook's eyes watered. | He ran out of onions. | He cut an onion. | cause | 1 | true | choice2 |
| The tree branch landed in the river. | The branch moved downstream. | The river's current became stronger. | effect | 0 | false | choice1 |
| The woman retired. | She received her pension. | She paid off her mortgage. | effect | 0 | false | choice1 |

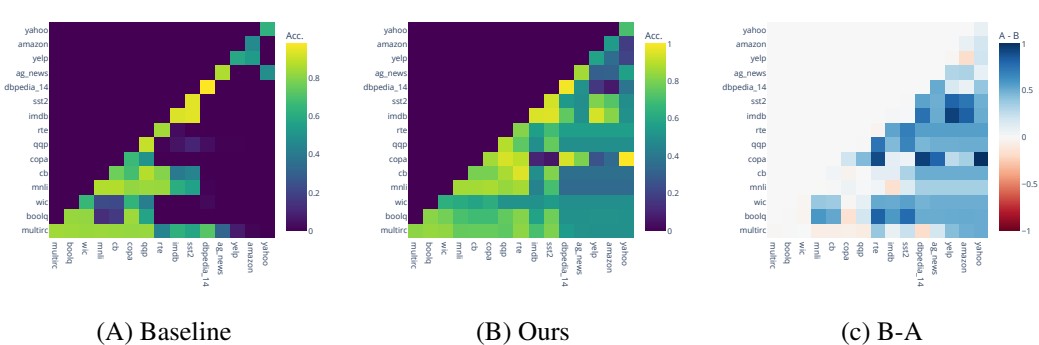

(A) Baseline          (B) Ours          (c) B-A

Figure 4: Heatmaps of backward transfer scores on previous tasks for Order L2. (A) shows results from the baseline (Progressive Prompts), (B) shows our method, and (C) presents the difference (B - A). Brighter values indicate better retention of earlier tasks.

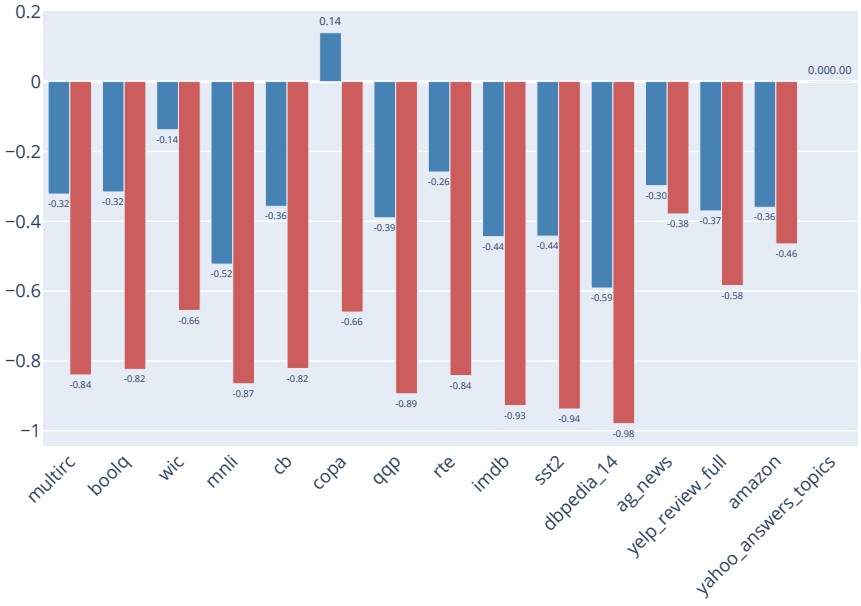

Figure 5: Per-task BWT comparison between our method (blue) and the baseline (red) for Order L2. Positive bars indicate improved retention of prior tasks.

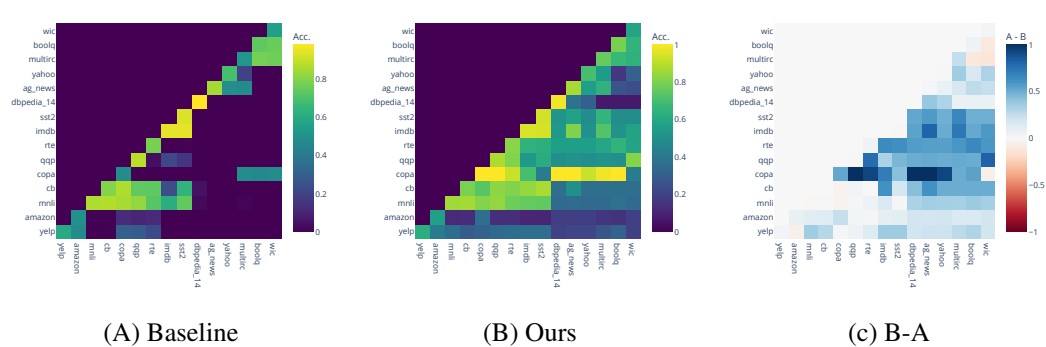

(A) Baseline    (B) Ours    (c) B-A

Figure 6: Heatmaps of backward transfer scores on previous tasks for Order L3. (A) shows results from the baseline (Progressive Prompts), (B) shows our method, and (C) presents the difference (B - A). Brighter values indicate better retention of earlier tasks.

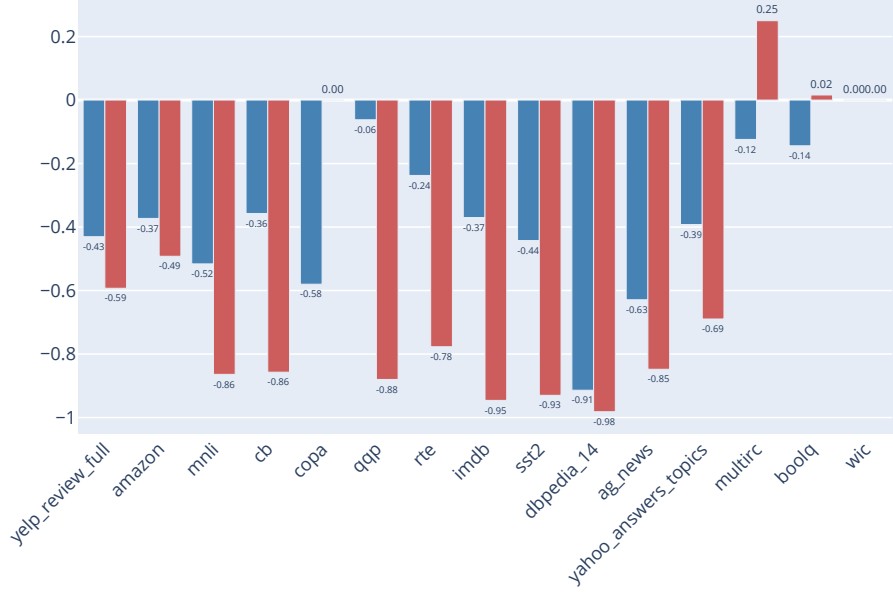

Figure 7: Per-task BWT comparison between our method (blue) and the baseline (red) for Order L3. Positive bars indicate improved retention of prior tasks.

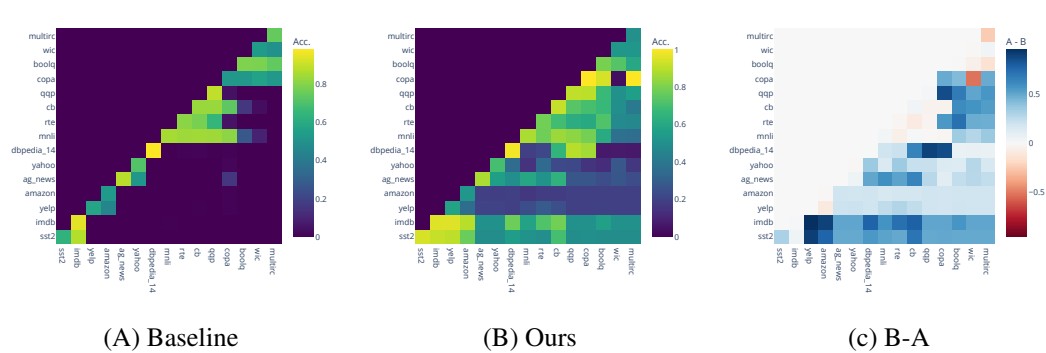

(A) Baseline                    (B) Ours                    (c) B-A

Figure 8: Heatmaps of backward transfer scores on previous tasks for Order L4. (A) shows results from the baseline (Progressive Prompts), (B) shows our method, and (C) presents the difference (B - A). Brighter values indicate better retention of earlier tasks.

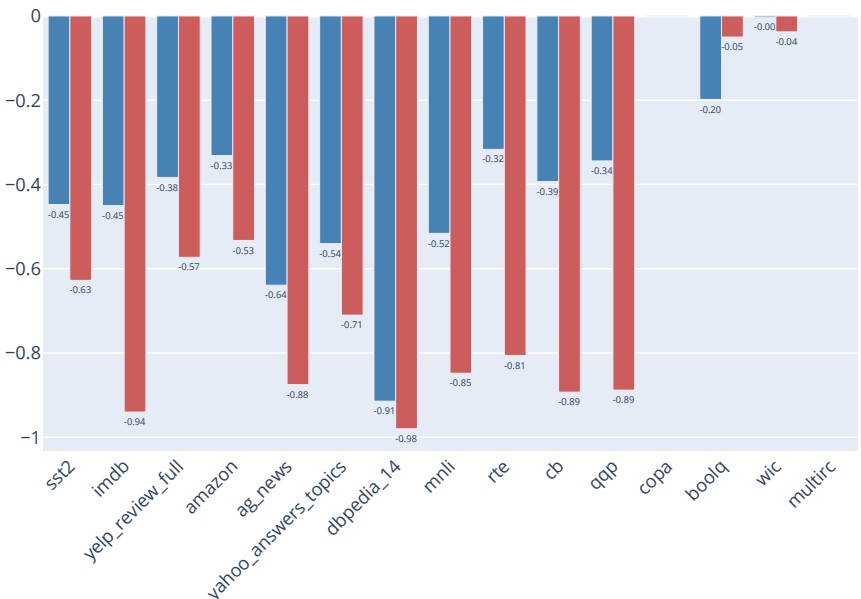

Figure 9: Per-task BWT comparison between our method (blue) and the baseline (red) for Order L4. Positive bars indicate improved retention of prior tasks.

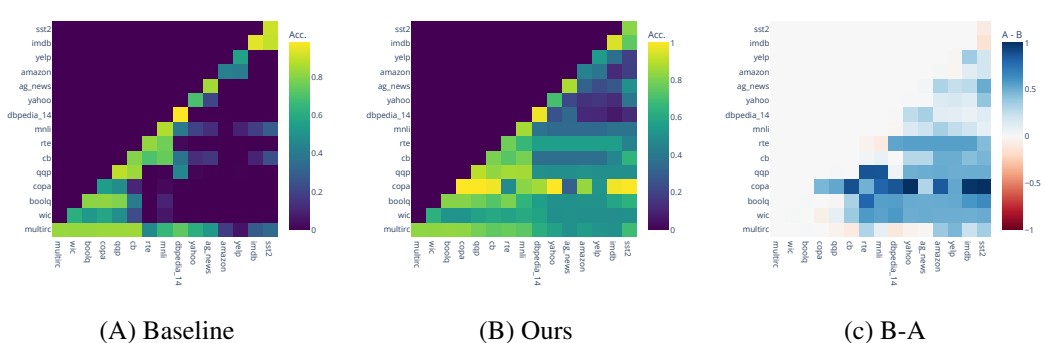

(A) Baseline           (B) Ours           (c) B-A

Figure 10: Heatmaps of backward transfer scores on previous tasks for Order L5. (A) shows results from the baseline (Progressive Prompts), (B) shows our method, and (C) presents the difference (B - A). Brighter values indicate better retention of earlier tasks.

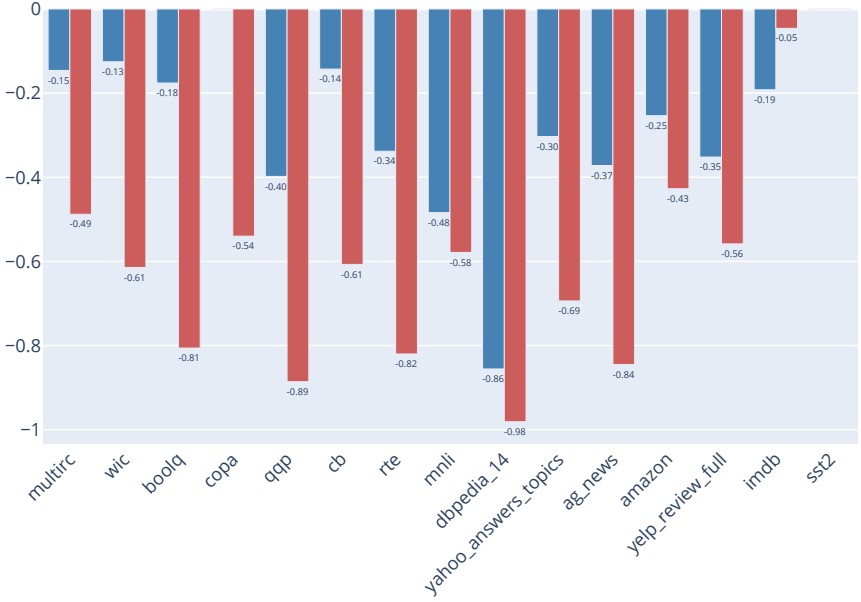

Figure 11: Per-task BWT comparison between our method (blue) and the baseline (red) for Order L5. Positive bars indicate improved retention of prior tasks.

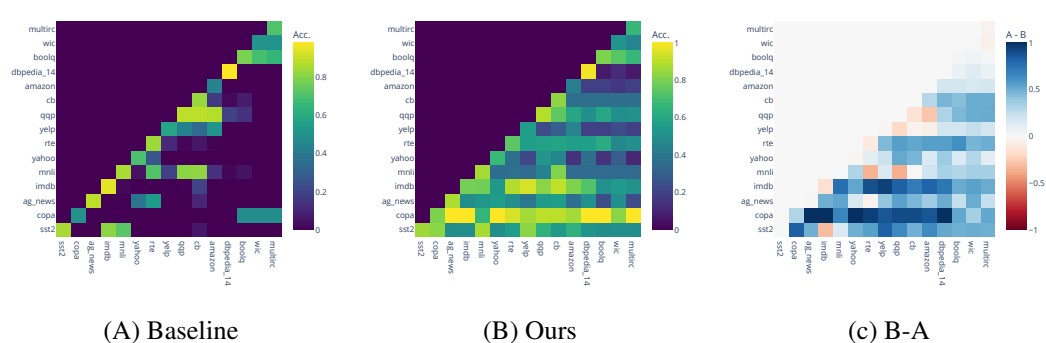

(A) Baseline          (B) Ours          (c) B-A

Figure 12: Heatmaps of backward transfer scores on previous tasks for Order L5. (A) shows results from the baseline (Progressive Prompts), (B) shows our method, and (C) presents the difference (B - A). Brighter values indicate better retention of earlier tasks.

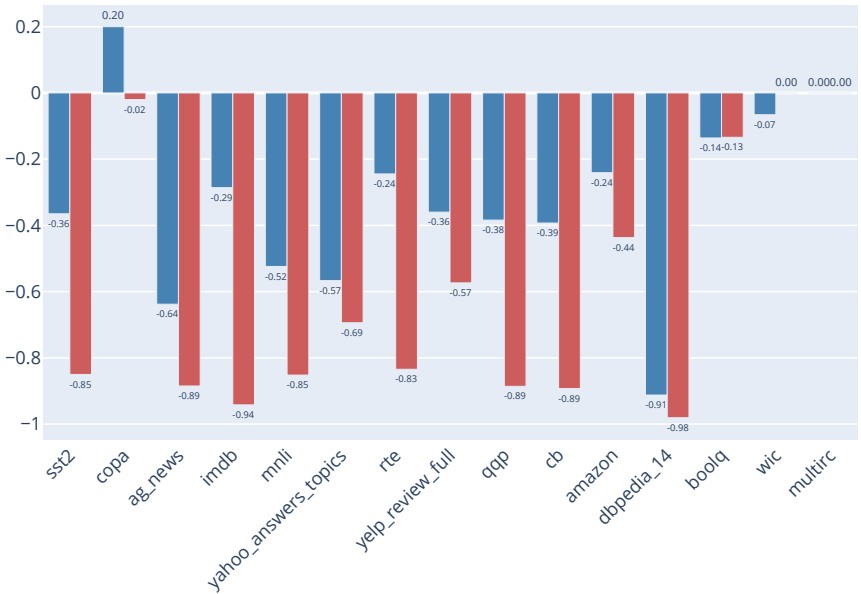

Figure 13: Per-task BWT comparison between our method (blue) and the baseline (red) for Order L6. Positive bars indicate improved retention of prior tasks.

