# OpenReview forum: "GRID: Scalable Task-Agnostic Prompt-Based Continual Learning for Language Models"
_ICLR.cc/2026/Conference — ICLR 2026 Conference Withdrawn Submission_

### Official Review · Reviewer_9iSF · 2025-10-26

**Soundness:** 3
**Presentation:** 2
**Contribution:** 2
**Rating:** 2
**Confidence:** 3

**Summary:**

This paper introduces a prompt-based continual learning method for preserving model performance on old tasks and optimizing the memory usage in task-agnostic scenarios.

**Strengths:**

1. This work tries to address the challenging task of continual learning in task-agnostic settings, achieving good results on long-sequence and negative transfer benchmarks.

2. The intuition of using gradient to determine the usefulness of prompts is interesting.

**Weaknesses:**

1. I am doubtful whether label mapping is still widely adopted in the mainstream decoder-only models. It looks to me that although the performance of decoder-only models may slightly underperform the encoder-decoder models, decoder-only models are scalable and do not require specific mapping strategies. Moreover, I am wondering why the authors do not apply thier method on decoder-only models such as Llama and Qwen.

2. There is redundency in writing. Both the introduction section and section 2.3 mention Lack of Task Awareness and Prompt Growth, which is unnecessary. Moreover, S4 and S5 in Figure 1 are not introduced in the paper.

3. The authors claim that their method reduces the memory storage. However, the calculation of the gradient requires more GPU memory than direct inference, which makes the claim not solid enough.

**Questions:**

1. Could you please provide some cases where the task id is not available during inference?

2. The authors mention a label set L_i in line 224. However, I do not see a related definition of L_i.

---

### Official Review · Reviewer_mf3C · 2025-10-31

**Soundness:** 3
**Presentation:** 3
**Contribution:** 2
**Rating:** 6
**Confidence:** 4

**Summary:**

This paper proposes GRID, a scalable and task-agnostic prompt-based continual learning framework for large language models. GRID enables task-agnostic inference by combining two key components: (1) a constrained decoding pipeline that leverages representative input sampling, automatic task identification, and restricted output spaces to mitigate label drift and latent forgetting; and (2) a gradient-guided prompt selection and compression mechanism that merges redundant prompts to ensure memory efficiency.

**Strengths:**

1. The paper addresses an underexplored but practically important setting where task identities are unknown at inference time. By integrating task identification and constrained decoding, GRID effectively mitigates label drift and latent forgetting—issues largely ignored in prior prompt-based continual learning works such as ProgPrompt and SHLPT.

2. The proposed gradient-guided prompt compression provides a simple yet effective mechanism to control prompt pool growth, reducing memory usage by over 60% without compromising performance. This contributes to the scalability and practical deployability of prompt-based continual learning in large language models.

**Weaknesses:**

1. **Lack of theoretical analysis.**
For example, the gradient-based prompt selection in Eq. (1–4) is heuristic: prompts with smaller gradient norms are merged by a simple weighted average. However, the paper provides no theoretical justification for why the gradient magnitude correlates with “informativeness”.

2. **Limited experimental evidence supporting the core claim on task-agnostic inference and constrained decoding.**
Although the paper claims that GRID enables task-agnostic continual learning through its constrained decoding mechanism, the experiments do not fully isolate or quantify this effect. For example, Tables 2–5 are conducted under fixed task sequences (L1–L6, NT1–NT3), where the model still implicitly follows task boundaries during training and testing. There is no experiment showing inference under truly mixed or unknown-task inputs, where samples from multiple tasks are presented without task labels.

**Questions:**

1. Could the authors provide more theoretical analysis to better justify the effectiveness of the gradient-based prompt selection mechanism?

2. Could the authors provide additional empirical evidence to further demonstrate the effectiveness of the constrained decoding mechanism, particularly in task-agnostic inference scenarios?

3. Have the authors explored or compared alternative dynamic selection or compression strategies beyond the gradient-based approach described in Equation (1)?

---

### Official Review · Reviewer_Zano · 2025-11-01

**Soundness:** 2
**Presentation:** 2
**Contribution:** 2
**Rating:** 2
**Confidence:** 3

**Summary:**

This paper proposes GRID which cleanly targets task-agnostic continual learning with a bounded prompt pool and offers a practical, well-engineered framework. The representative input sampling with task identification stabilizing decoding and constrained decoding curbs label drift and a gradient guided prompt selection cutting memory together improve prompt efficacy on downstream tasks. The authors also present ablations and runtime/memory. However, the conceptual novelty is modest as clustering, label remapping + constrained decoding, gradient scoring or prompt aggregation are widely known to everyone in this community.

**Strengths:**

The problem is very clear for task-agnostic inference with a bounded prompt pool.
The synergy of representative-input sampling + task ID and constrained decoding and gradient-guided prompt selection/merging. It cuts prompt memory $\approx \frac{2}{3}$ with better BWT/FTC than ProgPrompt/SHLPT, and includes ablations and runtime/memory reporting.

**Weaknesses:**

Useful engineering but the conceptual novelty is modest and the evaluation might be too simple.
Each component is well-known: representative sampling/clustering, label remapping + constrained decoding, and gradient-norm scoring. The “gradient-weighted merging” is a straightforward heuristic.
Even though the amount of tasks is large, most tasks are short-label classification (BoolQ, MNLI, SST-2, etc.). No open-ended generation, no tool use, no long-context or instruction-following streams, little domain/format shift, and few tasks where prompts truly interfere.
The threshold $\tau = \mu_{g} + \alpha \sigma_{g}$ is not analyzed.
Models are limited to T5/Flan-T5. The gains are not clear for modern chat LLMs or multimodals. Can the authors run larger and more recent models?

**Questions:**

Sensitivity to $\alpha$, cluster count, similarity cutoff, and representative-set size; show curves, not single points.
Any results on instruction-following models/tasks (e.g., FLAN mixtures, BIG-bench tasks), multi-label/structured outputs, multilingual, or tool-augmented tasks? Harder reasoning tasks? Long-form generation?
Are gains statistically significant across seeds? How many trials authors run? At least 5 trials should be repeated per setting for statistical significance.

---

### Official Review · Reviewer_t1ow · 2025-11-02

**Soundness:** 2
**Presentation:** 3
**Contribution:** 2
**Rating:** 2
**Confidence:** 4

**Summary:**

GRID is a prompt-based continual learning framework tackling two key challenges: severe performance degradation under task-agnostic inference and prompt memory accumulation as task sequences grow. It introduces a novel decoding mechanism—leveraging automatic task identification and constrained decoding—to enhance backward knowledge transfer. Additionally, GRID uses a gradient-guided prompt selection strategy that merges less informative prompts into a single representation for scalable, memory-efficient learning. Experiments on long-sequence and negative transfer benchmarks show improved average accuracy and backward transfer, competitive forward transfer, and substantially reduced prompt memory usage.

**Strengths:**

1. The problem is interesting, there are a lot of literature for the CL in the visual domain but limited work in the NLP domain, this work maybe beneficial for the NLP community.

2. The proposed architecture seems novel, gradient norm based prompt pool selection looks interesting. The gradient-guided prompt selection and merging to prevent unbounded prompt growth, maintaining high performance with over 60% reduced memory compared to prior prompt-based continual learners.

3. Paper does not require the task id during inference, it follow CIL setting

4. GRID achieves superior accuracy with fewer parameters, no rehearsal data, and stable performance, offering a unified, scalable, and task-agnostic framework for real-world continual learning in NLP.

**Weaknesses:**

1. In the vision domain, the CIL setting is very common for the prompt based continual learning, here author highlights its key challenge, why? On the high level once tokenization is done both the language and vision model are similar, then why same concept [a,b,c] can not be applied here?

2. The task identification method in section 3.1 looks violates the CL setting, here paper use the pretrained (say Phi) model for the task id prediction, which is much powerful and mostly it know all the sequence of task earlier.

3. The result are shown on the T5 architecture only and over a single dataset which is not sufficient to defend the proposed model. I think other recent model and diverse dataset are needed to validate the proposed approach robustness and performance.

4. The model is not robust to the task order, which is a challenge to the proposed model.

**References**

[a] Coda-prompt: Continual decomposed attention-based prompting for rehearsal-free continual learning, CVPR-23
[b] Dualprompt: Complementary prompting for rehearsal-free continual learning, ECCV-22
[c] Convolutional Prompting meets Language Models for Continual Learning, CVPR-24

**Questions:**

The problem addressed in the paper is interesting; however, the evaluation is limited, making it difficult to comprehensively assess the model’s effectiveness.

Additionally, recent vision-based continual learning models employing prompt-based techniques implicitly select suitable prompts from a prompt pool without explicit task supervision. It remains unclear why achieving similar task-agnostic prompt selection poses a greater challenge in the NLP domain.

---

### Note · Authors · 2025-11-12

I have read and agree with the venue's withdrawal policy on behalf of myself and my co-authors.